# Automated Final Lesion Segmentation in Posterior Circulation Acute Ischemic Stroke Using Deep Learning

**DOI:** 10.3390/diagnostics11091621

**Published:** 2021-09-04

**Authors:** Riaan Zoetmulder, Praneeta R. Konduri, Iris V. Obdeijn, Efstratios Gavves, Ivana Išgum, Charles B.L.M. Majoie, Diederik W.J. Dippel, Yvo B.W.E.M. Roos, Mayank Goyal, Peter J. Mitchell, Bruce C. V. Campbell, Demetrius K. Lopes, Gernot Reimann, Tudor G. Jovin, Jeffrey L. Saver, Keith W. Muir, Phil White, Serge Bracard, Bailiang Chen, Scott Brown, Wouter J. Schonewille, Erik van der Hoeven, Volker Puetz, Henk A. Marquering

**Affiliations:** 1Department of Biomedical Engineering and Physics, Amsterdam UMC, Location AMC, 1105 Amsterdam, The Netherlands; r.zoetmulder@amsterdamumc.nl (R.Z.); p.r.konduri@amsterdamumc.nl (P.R.K.); i.v.obdeijn@student.utwente.nl (I.V.O.); i.isgum@amsterdamumc.nl (I.I.); 2Department of Radiology and Nuclear Medicine, Amsterdam UMC, Location AMC, 1105 Amsterdam, The Netherlands; c.b.Majoie@amsterdamumc.nl; 3Informatics Institute, University of Amsterdam, 1097 Amsterdam, The Netherlands; E.Gavves@uva.nl; 4Department of Neurology, Erasmus MC University Medical Center, 3015 Rotterdam, The Netherlands; d.dippel@erasmusmc.nl; 5Department of Neurology, Amsterdam UMC, Location AMC, 1105 Amsterdam, The Netherlands; y.b.roos@amsterdamumc.nl; 6Radiology, Foothills Medical Centre, University of Calgary, Calgary, AB T2N 2T9, Canada; mgoyal@ucalgary.ca; 7Department of Clinical Neurosciences, Hotchkiss Brain Institute, University of Calgary, Calgary, AB T2N 4N1, Canada; 8Department of Radiology, The University of Melbourne & The Royal Melbourne Hospital, Melbourne, VIC 3050, Australia; Peter.Mitchell@mh.org.au; 9Department of Medicine and Neurology, Melbourne Brain Centre at the Royal Melbourne Hospital, University of Melbourne, Melbourne, VIC 3052, Australia; dr.bruce.campbell@gmail.com; 10Department of Neurological Surgery, Rush University Medical Center, Chicago, IL 60612, USA; brainaneurysm@mac.com; 11Department of Neurology, Community Hospital Klinikum Dortmund, 44137 Dortmund, Germany; Gernot.Rudel@klinikumdo.de; 12Cooper Neurological Institute, Cooper University Medical Center, Camden, NJ 08103, USA; tudorjovin@gmail.com; 13Department of Neurology and Comprehensive Stroke Center, David Geffen School of Medicine at UCLA, Los Angeles, CA 90095, USA; jsaver@mednet.ucla.edu; 14Institute of Neuroscience and Psychology, University of Glasgow, Glasgow G12 8QB, UK; keith.muir@glasgow.ac.uk; 15Translational and Clinical Research Institute, Faculty of Medical Sciences, Newcastle University, Newcastle upon Tyne NE1 7RU, UK; phil.white@newcastle.ac.uk; 16Department of Neuroradiology, Newcastle upon Tyne Hospitals, Newcastle upon Tyne NE1 4LP, UK; 17INSERM U1254, IADI, University Hospital, Neuroradiology, 54511 Nancy, France; s.bracard@chru-nancy.fr; 18INSERM CIC-IT 1433, University Hospital, 54511 Nancy, France; B.CHEN@chru-nancy.fr; 19Altair Biostatistics, St Louis Park, MN 55416, USA; b_scott_brown@yahoo.com; 20St. Antonius Hospital, 3435 Nieuwegein, The Netherlands; w.schonewille@antoniusziekenhuis.nl; 21Department of Radiology, St. Antonius Hospital, P.O. Box 2500, 3430 Nieuwegein, The Netherlands; e.van.der.hoeven@antoniusziekenhuis.nl; 22Department of Neurology, Dresden University Stroke Centre, Technical University Dresden, Fetscherstraße 74, 01307 Dresden, Germany; volker.puetz@neuro.med.tu-dresden.de

**Keywords:** posterior stroke, segmentation, transfer learning, deep learning, CT

## Abstract

Final lesion volume (FLV) is a surrogate outcome measure in anterior circulation stroke (ACS). In posterior circulation stroke (PCS), this relation is plausibly understudied due to a lack of methods that automatically quantify FLV. The applicability of deep learning approaches to PCS is limited due to its lower incidence compared to ACS. We evaluated strategies to develop a convolutional neural network (CNN) for PCS lesion segmentation by using image data from both ACS and PCS patients. We included follow-up non-contrast computed tomography scans of 1018 patients with ACS and 107 patients with PCS. To assess whether an ACS lesion segmentation generalizes to PCS, a CNN was trained on ACS data (ACS-CNN). Second, to evaluate the performance of only including PCS patients, a CNN was trained on PCS data. Third, to evaluate the performance when combining the datasets, a CNN was trained on both datasets. Finally, to evaluate the performance of transfer learning, the ACS-CNN was fine-tuned using PCS patients. The transfer learning strategy outperformed the other strategies in volume agreement with an intra-class correlation of 0.88 (95% CI: 0.83–0.92) vs. 0.55 to 0.83 and a lesion detection rate of 87% vs. 41–77 for the other strategies. Hence, transfer learning improved the FLV quantification and detection rate of PCS lesions compared to the other strategies.

## 1. Introduction

Infarct volume, representing the tissue damage after an acute ischemic stroke (AIS), is commonly considered as a surrogate endpoint for the primary functional outcome (modified Rankin Scale (mRS) after 90 days) [1]. Various trials have shown a strong association of final lesion volume (FLV) with functional outcome in patients suffering from a stroke due to a large vessel occlusion in the anterior circulation [1,2].

However, in patients with a stroke due to a posterior circulation stroke (PCS), the relation between FLV and outcome is understudied [3]. The low number of studies addressing this relation may be due to the combination of two reasons; the relatively low prevalence of PCS compared to anterior circulation stroke (ACS) and the lack of automated analysis of PCS lesion volume assessment.

With the huge effectiveness of endovascular treatment of anterior circulation stroke patients, treatment of posterior stroke has attained renewed interest in various studies and trials. For example, the recently completed BASICS trial [4] could not show a beneficial effect of endovascular treatment with functional outcome used as an outcome measure. Alternatively, secondary outcome measures such as FLV might show a beneficial effect of certain treatments since functional outcome, as addressed by the mRS, is a rather coarse outcome measure, which is also affected by many other confounders [5]. Developing methods that automatically segment lesions due to a posterior circulation stroke (PCS) would help investigate FLV as a surrogate outcome for this type of stroke.

Solutions for the automatic segmentation of FLV based on convolutional neural networks (CNNs) have been presented in the literature for CT and MR imaging [6,7]. However, these studies have only considered the FLV of patients with an AIS due to an occlusion of the anterior circulation [1,6]. To achieve good performance, CNNs typically require large amounts of labeled training data. However, PCS constitutes only 26% of AIS cases [8,9], and thus, training of CNNs for automatic PCS lesion segmentation is hindered by the limited availability of data. Furthermore, the applicability of methods developed for ACS FLV segmentation on posterior stroke lesion segmentation is unknown.

Several methods exist for dealing with a lack of data to train a CNN. One method that reduces the data needed to train CNNs by reusing knowledge is transfer learning [10]. To perform transfer learning, a CNN is pre-trained on a task for which large amounts of image data are available and fine-tuned on a different task for which little image data are available. Transfer learning has been successfully applied to solve various medical image analysis problems [11,12,13,14].

We evaluate strategies to create automated PCS lesion segmentation by using image data from patients with ACS and patients with PCS. We hypothesize that transfer learning utilizing data of ACS lesions improves automatic PCS lesion segmentation performance compared to alternative strategies: training a CNN on only ACS lesions, only on PCS lesions, or on the combination of ACS and PCS lesions.

## 2. Materials and Methods

### 2.1. Patient and Image Data

All involved patients in this retrospective study or their legal representatives provided written informed consent. The medical ethics committee of each participating hospital approved the use of the data after anonymization.

The Hermes dataset consists of 1665 patients who suffered from an ACS and was obtained from the HERMES collaboration [15], which investigated the effectiveness of endovascular therapy for treating ACS. This collaboration combined data from seven clinical randomized trials and collected data between December 2010 and December 2014. The scans contain a varying number of slices, which each have 512 rows and columns. The inclusion criteria are shown in Figure A1A. Patients were excluded if no follow-up non-contrast computed tomography (FU-NCCT) was made in the time window between 12 h and 2 weeks after stroke onset or if the preprocessing steps were unsuccessful. In total, 1018 patients out of the 1665 patients were included. Baseline characteristics of the included patients are shown in Table 1. 

The BASICS dataset consists of 168 patients who suffered from a PCS and was obtained from the BASICS trial [16,17], which investigated the effectiveness of endovascular therapy for treating patients with a PCS. This trial included patients from 23 centers, and the data were collected between 2011 and 2019. The scans are composed of a varying number of slices that each have 512 rows and columns. Imaging parameters are shown in Table A1. Inclusion criteria for our study are shown in Figure A1B.

Infarcts evolve over time, and the most recent FU-NCCT scans imaged the most evolved infarcts with the lowest densities. As a result of the more distinct presence of the infarcts in later scans, these scans are commonly used for the infarct volume assessment in clinical practice and clinical research. Hence, the latest FU-NCCT scan was used if multiple scans were available for the same patient. Patients were excluded if no FU-NCCT was made or if the follow-up image was of insufficient quality.

In total, 107 patients out of the 168 available patients were included. The mean and median time between symptom onset and the most recent follow-up NCCT are respectively, 98.5 h and 28 (IQR: 24–32). Baseline characteristics of the included BASICS patients are shown in Table 1. The infarcted regions, as expressed in affected PC-ASPECTS regions, are shown in Table A2 in the Appendix A.

### 2.2. Reference Segmentations

For patients with an ACS, reference segmentations were obtained by manual annotation by one of two experienced observers on the most recent FU-NCCT. The annotation procedure is outlined in [1]. In summary, a window width of 30 Hounsfield Units (HU) and a center level of 35 HU was set in ITK-Snap [18]. All hypo-dense regions on the ipsilateral hemisphere including edema were included in the segmentations. Infarcted tissue in the ipsilateral hemisphere with signs of an old infarct were excluded from the reference segmentations. Parenchymal hemorrhages adjacent to or within the affected area were included in the reference segmentations. Finally, reference segmentations were checked and, if necessary, corrected by one of three radiologists, each of whom had more than 5 years of experience.

Reference segmentations of lesions caused by a PCS on FU-NCCT scans were manually created by a single trained observer (IVO) and were checked by an experienced radiologist (CBLMM) who has more than 15 years of experience. Lesions were segmented by using the aforementioned window width and center level using ITK-Snap software [18]. PC-ASPECTS scores [19], which were scored by two experienced radiologists, were used when available to identify the infarcted territory.

### 2.3. Preprocessing

The intracranial region as a volume of interest was obtained automatically using a combination of preprocessing steps [6]. The bone was segmented using a threshold-based segmentation by selecting all voxels with an intensity of 170 HU or higher. Subsequently, the foramina, except the foramen magnum, were closed using morphological filters, and a region-growing algorithm was applied to select the intracranial volume. To obtain the final volume of interest, the region caudal to the foramen magnum was excluded.

To ensure the same size, orientation, and voxel sizes, all scans were aligned by automatically registering the images to a common space using rigid and affine transformations. Images were registered using the Mattes Mutual Information [20] with a gradient descent optimizer. In addition to registration, the scans were downsampled to allow the entire scan to be passed into the CNNs. After the preprocessing, each scan had a size of 256 × 256 × 32, with a slice thickness of 5 mm.

The voxel intensities were clipped between −20 and 120 HU and subsequently normalized between minus one and one. The preprocessing was done using SimpleITK [21,22] and Python 2.7.

### 2.4. CNN for Automatic Posterior Circulation Lesion Segmentation

The preprocessed images were input to a CNN which consisted of three-dimensional convolutional kernels. The architecture of the CNN (shown in Figure A2A in the Appendix A) was inspired by U-Net [23] and ResNet [24]. The CNN consisted of a downsampling path and an upsampling path. The downsampling path started with an input block, consisting of a convolution with a kernel size of three and a stride of one, which was followed by a max pooling layer with a pooling size of two and a stride of two. The dimensions of the input were 256 × 265 × 32, and the input block generated features that consisted of eight channels. 

Subsequent to the input block, three downsampling blocks were added, consisting of three 3D ResNet layers, as shown in Figure A2B in the Appendix A. The first two blocks were followed by average pooling with a stride and pooling size of two. 

The upsampling path started with a transposed convolution using a stride of two. Next, two upsampling blocks followed by an output layer were added. Each upsampling block consisted of two ResNet layers, which were followed by a transposed convolution with a stride of two and a kernel size of three. Each upsampling block took the featu res from the previous block and the corresponding downsampling block and concatenated them. The output block consisted of two ResNet blocks followed by a convolutional layer. The CNNs were implemented using Tensorflow 1.5.

### 2.5. Experimental Setup

Four different training strategies for CNNs were evaluated: A CNN was randomly initialized and trained on images of patients in the HERMES dataset (ACS-CNN), BASICS dataset (PCS-CNN), and the HERMES and BASICS datasets combined (CD-CNN). The ACS-CNN was used to establish the generalization ability of a CNN trained on ACS to PCS lesion segmentation. The PCS-CNN served as a baseline for training with limited but representative data. The CD-CNN was used as a benchmark if both ACS and PCS data were available, but no transfer learning was used. Transfer learning reused the weights from the trained ACS-CNN to initialize all but the last block of the CNN, and it fine-tuned by updating all the weights, using images from the BASICS dataset (TL-CNN). 

The CNNs used group normalization with four groups, the Leaky ReLU activation function, a batch size of two, and the Adam optimizer. The loss function used was the weighted binary cross-entropy. The initial learning rate for the ACS-CNN and CD-CNN was 10^−3^ and was decayed stepwise after 5, 10, 15, and 20 epochs to respectively, 5 × 10^−4^, 2 × 10^−4^, 10^−4^, and 10^−5^. These networks were trained for 25 epochs. The initial learning for the PCS-CNN and TL-CNN was 10^−5^ and was decayed after 25, 50, and 75 epochs to respectively, 5 × 10^−6^, 2 × 10^−6^, and 10^−6^. These networks were trained and fine-tuned for 100 epochs. The weight decay was set to 10^−5^.

Data augmentation was applied at training time. The images were rotated at a randomly chosen angle between zero and ten degrees along the axial plane in either direction or were randomly flipped along the sagittal plane.

We evaluated the performance of the ACS-CNN to check whether the model converges during pre-training. The ACS dataset was split randomly into scans for training (85%), validation (5%), and testing (10%). For this approach, we used the entire PCS dataset for evaluation. Thus, we used stratified 5-fold cross-validation. Given the number of available PCS patients, the first four testing splits consisted of 20% and the fifth consisted of 22% of the data. The training splits were of equal size and consisted of 78% of the PCS data. 

### 2.6. Evaluation

The reliability between the automatically and manually segmented volumes was evaluated with the intraclass correlation coefficient (ICC) including the 95% Confidence Interval (95% CI). The ICC was interpreted in accordance to the American Psychological Association [25]. Following their guidelines, an ICC < 0.4 is defined as poor, an ICC between 0.4 and 0.6 is defined as fair, an ICC between 0.6 and 0.75 is defined as good, and an ICC greater than 0.75 is defined as excellent. In addition, a Bland–Altman analysis was performed to assess the bias and limits of agreement (LoA) in volume measurements. Statistical significance between ICCs was evaluated by using Fisher’s *r*-to-*z* transformation.

We determined whether our model accurately detected lesions independent of size. Thus, we calculated the ratio of the total number of correctly detected lesions and the total number of lesions as determined by the ground truth in the dataset. A lesion was defined as detected if the percentage of overlapping voxels between the automatic and reference segmentations was larger than a predefined minimum. However, in case of small thresholds, a non-zero overlap of automated and reference lesion segmentation could be caused by chance. To account for this issue, the required minimum percentage of overlapping voxels to count a lesion as detected was set to be either greater than zero percent or greater than a more conservative 20%. Next, the effect of lesion volume on the lesion detection rate was studied by excluding lesions of a progressively larger volume in the reference segmentations. This latter cutoff was set between zero and 4 mL with increments of 0.5 mL.

The segmentation performance of the automatic methods was evaluated by calculating the Dice coefficient as an overlap measure between the reference and the automatic segmentation. Normality of the distribution of the Dice coefficients was assessed before pairwise statistical testing by using the Shapiro–Wilk test. If the Dice coefficients were normally distributed, a paired t-test was used; otherwise, a Wilcoxon rank sum test was used. *p*-values were corrected for the family wise error rate using the Bonferroni correction. All statistical testing was done using the python library Pingouin, v.0.3.1 [26] (University of California, Berkeley, CA, USA).

## 3. Results

Baseline characteristics of the patients in BASICS and HERMES datasets were compared. Patients in the HERMES dataset had a similar age to patients in the BASICS dataset. Diabetes (26.2% vs. 16.6%, *p* < 0.05) and prior stroke (19.6% vs. 11.9%, *p* < 0.05) occurred more frequently in patients in the BASICS dataset. However, atrial fibrillation (12.1% vs. 30.8%, *p* < 0.01) occurred more frequently in patients in the HERMES dataset. 

The median FLV in patients with PCS was 11 (IQR: 3.4–36) mL. The ICCs for volume assessments for the TL-CNN, PCS-CNN, CD-CNN, and ACS-CNN were 0.88 (95% CI: 0.83–0.92), 0.80 (95% CI: 0.72–0.86), 0.83 (95% CI: 0.76–0.88), and 0.55 (95% CI: 0.4–0.67), respectively. The ICC of the TL-CNN was significantly larger than the ICCs of the ACS-CNN (*p* < 0.01) and PCS-CNN (*p* = 0.02). The ICC of the ACS-CNN was significantly smaller than the ICCs of the PCS-CNN (*p* < 0.01) and CD-CNN (*p* < 0.01). In addition, the Bland–Altman analysis for each of the CNNs resulted in biases ranging from 0.8 mL for the TL-CNN to 13.5 mL for the ACS-CNN. The LoAs were the smallest for the TL-CNN with −29 to 30 mL and largest for the ACS-CNN with −32 to 59 mL. The bias and LoAs of the volume measurements are shown in Figure 1 and, Table A3 in the Appendix A.

The lesion detection rate of the TL-CNN was higher than for the other learning strategies (Figure 2), which shows that the lesion detection rate increases with increasing lesion volume and that the lesion detection rate decreases with increasing thresholds of overlapping voxels.

The TL-CNN, PCS-CNN, CD-CNN, and ACS-CNN achieved a Dice coefficient of 0.25 ± 0.08, 0.21 ± 0.06, 0.16 ± 0.06, and 0.07 ± 0.03, on the overall PCS test set, respectively (Figure 3). The Dice coefficients were not normally distributed. Hence, a Wilcoxon rank sum test was used. Results of the Wilcoxon rank sum test are shown in Table A4. For the anterior circulation stroke lesions, the ACS-CNN achieved an average Dice coefficient of 0.60 ± 0.07.

## 4. Discussion

Our study found that transfer learning results in a high level of agreement between manually delineated and automatically quantified lesion volumes on the follow-up NCCTs of patients with a PCS. Furthermore, we found that transfer learning resulted in higher spatial accuracy and larger volume agreement of automatic PCS lesion segmentation compared to the other strategies. In addition, the TL-CNN models also detected a larger number of PCS lesions in comparison to the other strategies. Moreover, our results indicate that the ACS-CNN models, which were trained on only patients who suffered from an ACS, do not generalize to PCS lesion segmentation. 

Our study is the first to address the automated segmentation of posterior circulation stroke lesions on sub-acute follow-up NCCT. Previous work include anterior stroke lesion segmentation in a variety of imaging modalities, such as baseline CTP [27], baseline CTA [28], follow-up NCCT [6], and baseline and follow-up DWI [29] using multiple approaches and addressing various types of stroke.

A previous study focused on developing a CNN-based method for automatic ACS lesion segmentation on FU-NCCT [6] with a higher spatial overlap accuracy than found in our study. This could be explained by the larger lesion volumes in their population (median FLV of 48 vs. 11 mL) and the larger dataset available for training. Furthermore, beam-hardening artifacts are common in the posterior fossa on NCCT. Beam-hardening artifacts may obscure lesions caused by a posterior circulation stroke, making these lesions harder to detect than lesions caused by anterior circulation stroke. Finally, the Dice coefficient is a global overlap metric, which is valuable when comparing large delineations. If the objects of interest are smaller, such as for the lesions in our population, the Dice coefficient may be too sensitive to small errors. Therefore, the Dice coefficient must be interpreted in addition to other metrics, such as the lesion detection rate and the correspondence in lesion volume, as expressed with the ICC between the automatically quantified and reference volumes, and not in isolation. The latter measure, lesion volume, is also clinically a more relevant measure than the spatial overlap. If we put the performance of our algorithms in context using the ICC, it shows that our algorithm can adequately predict FLV, which has a strong association with functional outcome in clinical practice. In addition, the bias and LoAs of the FLV were comparable to other studies assessing infarct volumes for similar problems. One study found a bias between the automatically and manually segmented FLV on FU-NCCT of ACS patients with subtle hypo-densities of 29 mL with LoA of −91 to 149 mL, which are both considerably larger compared to our study [6]. Another study found a bias between the automatically and manually segmented FLV on baseline NCCT of ACS patients of 11 mL with LoA of −59 to 80 mL [30]. Our algorithm, which utilized transfer learning, resulted in a bias of 0.84 mL and LoA of −29 to 30 mL, showing that, on average, the bias in FLV is small. A weakness, as in many other studies, is the large LoA values in our study. Since the infarct volumes are clinically mostly used in prognostic models, it is unknown to what extent this patient-specific variation in FIV influences prognosis. However, the large LoAs suggest that patient-specific analysis still has to be improved before the algorithm could be applied in a clinical setting.

Other methods for automatic ACS lesion segmentation used information from the contralateral hemisphere to improve segmentation accuracy [28,31]. PCS lesions can affect both hemispheres; hence, comparing information between the ipsilateral and its contralateral hemisphere is unlikely to improve the accuracy.

Automated stroke lesion segmentation has also been developed for chronic stroke lesions on T1 MRI. Chronic stroke lesions have been segmented by using a random forest classifier to segment lesions in the left hemisphere by using hand-crafted image features [32]. Another study used a deep residual network to segment lesions on images of the ATLAS dataset, which contains manually traced lesions on 304 T1-weighted MRI images [33]. Both studies achieved higher similarity scores than our method, which could be explained by the larger FLVs and higher sensitivity provided by T1-weighted MRI images.

In previous research, transfer learning has also been successfully applied to improve the accuracy of various other medical image segmentation tasks. One study used CNNs pre-trained on eight different medical image segmentation tasks on various imaging modalities to improve automatic lung, liver, and liver tumor segmentation [34]. Unlike the aforementioned study, our study pre-trained on a single imaging modality and task. Another study pre-trained CNNs using self-supervised tasks to improve lung nodule, liver, and brain tumor segmentation [35]. In agreement with our study, the results of prior work indicate that for medical image segmentation, transfer learning can be beneficial. 

Other work using transfer learning for medical image tasks included CNNs pre-trained on ImageNet [36] as a benchmark. These studies used transfer learning to improve the performance on medical image analysis tasks on 2D images. However, using ImageNet for transfer learning was less likely to be suitable for our study, because prior work has shown that ImageNet pre-training improves performance on medical image analysis tasks less than using a pre-trained 3D model [35].

This study has several limitations. First, most of the FU-NCCT scans of PCS patients included in this study were obtained 24 h after the onset of AIS. Creating manual reference segmentations of the lesions on these early FU-NCCT scans is more challenging owing to the subtle differences in HU values after 24 h. In addition, FLV segmentations for patients with PCS were performed by only one trained observer who used PC-ASPECTS scores to verify the lesion location and was supervised by a radiologist (CBLMM). Since the assessment was only performed by a single observer, inter-observer agreement of the reference standard could not be assessed. Second, this study suffered from a low number of available PCS patients. Therefore, an even lower number of patients would be included if the data were divided into training, validation, and test sets. This would have lowered the generalizability of the results to the PCS patient population. To overcome this, five-fold cross-validation was used to allow all the data to be used as testing data in the analysis and to assess the stability of the presented results. Third, the CNNs were not accurate at detecting lesions with a volume smaller than 2 mL. If a patient is suspected of having lesions with a small volume, results from the presented algorithm should be verified by an expert evaluation. Transfer learning allowed the CNNs to reuse information learned from ACS lesion segmentation to segment lesions caused by PCS. The resulting improvement in PCS lesion segmentation is likely due to the similarity between the pre-training and the fine-tuning tasks [37]. However, in our approach, the detection and segmentation of small lesions and the segmentation of lesions that are connected to cerebrospinal fluid-filled areas is still suboptimal. Fourth, it is quite likely that the swelling due to stroke has caused herniation in multiple patients. In our study, we did not exclude patients based on herniation, also because CT cannot distinguish infarcts resulting from the acute stroke versus herniation.

Deep learning is potentially valuable for automating demanding tasks in the quantification of radiological imaging. It is well-known that deep learning requires large amounts of data to train algorithms, which may suggest a limited applicability of deep learning in less common diseases. This study also shows that deep learning models that are trained on a more general, less specific disease may not be sufficient. Here, we presented an alternative approach based on transfer learning and showed that deep learning models can be pre-trained on similar diseases and fine-tuned on the specific rarer disease. 

To conclude, the presented transfer learning approach improves the automatic detection and segmentation of posterior circulation stroke lesions compared to the evaluated commonly used training strategies. The presented automated posterior stroke lesion segmentation method allows the inclusion of lesion volume as an image outcome measure and as a metric to predict outcome in large-scale clinical trials and potentially as a first step toward clinical application.

## Figures and Tables

**Figure 1 diagnostics-11-01621-f001:**
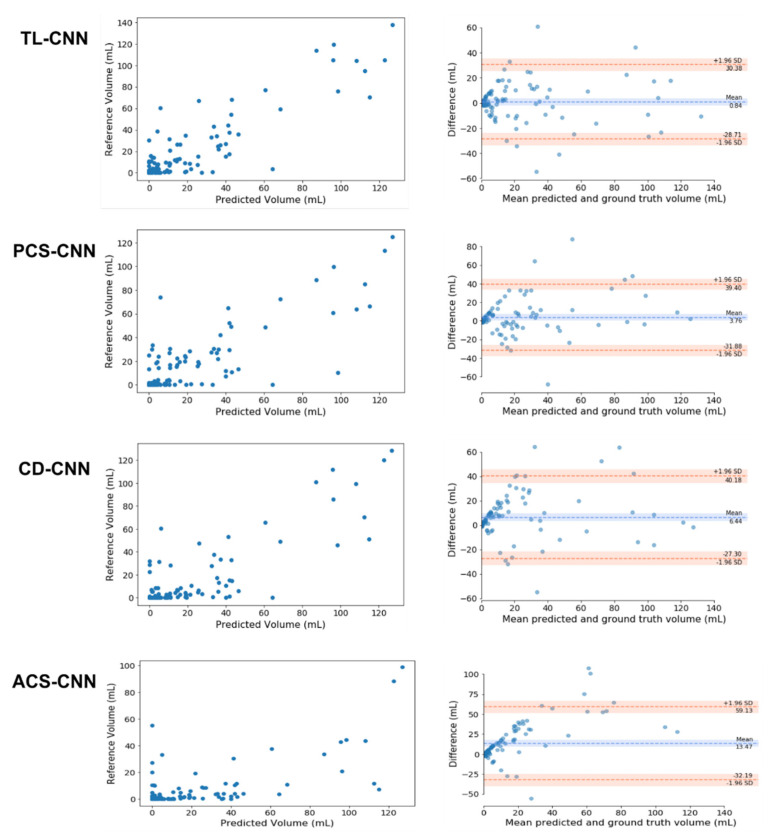
Comparison of the automated and reference segmentation volume for the Transfer Learned CNN (TL-CNN), Posterior Circulation Stroke CNN (PCS-CNN), Combined Datasets CNN (CD-CNN), and Anterior Circulation Stroke CNN (ACS-CNN). Left column: Scatter plots comparing lesion volumes derived from the reference segmentations (*y*-axis) and from the automatic segmentations determined by the CNN (*x*-axis). Right column: Bland–Altman plots of the lesion volumes. The volumes corresponding to the reference and automatic segmentations are shown on the *x*-axis and the volume difference is shown on the *y*-axis.

**Figure 2 diagnostics-11-01621-f002:**
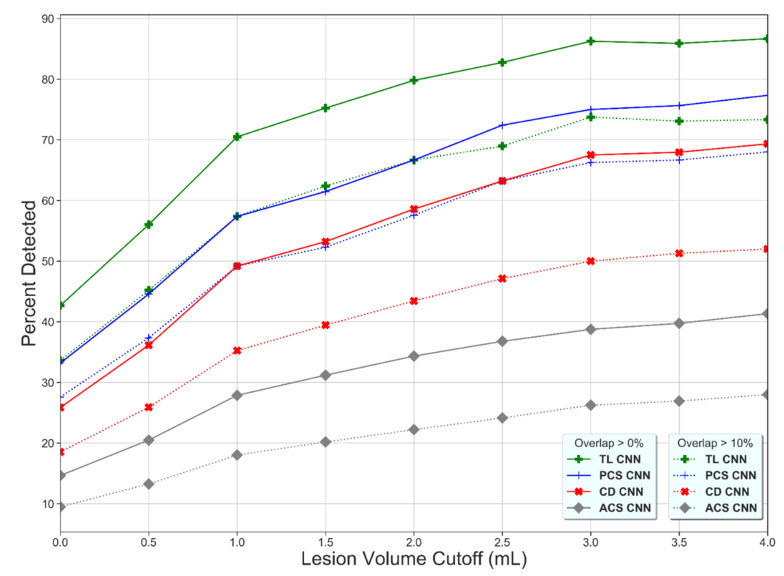
Percentage of detected lesions (*y*-axis) as a function of the minimum volume requirement (*x*-axis) and the minimum percentage of overlapping voxels for the Transfer Learned CNN (TL-CNN), Posterior Circulation Stroke CNN (PCS-CNN), Combined Datasets CNN (CD-CNN), and Anterior Circulation Stroke CNN (ACS-CNN) (green, blue, red, and gray lines). For all methods, a higher lesion volume cutoff results in a higher percentage of detected lesions. The lower overlapping voxel requirement, the higher the percentage of detected lesions (dotted versus solid lines).

**Figure 3 diagnostics-11-01621-f003:**
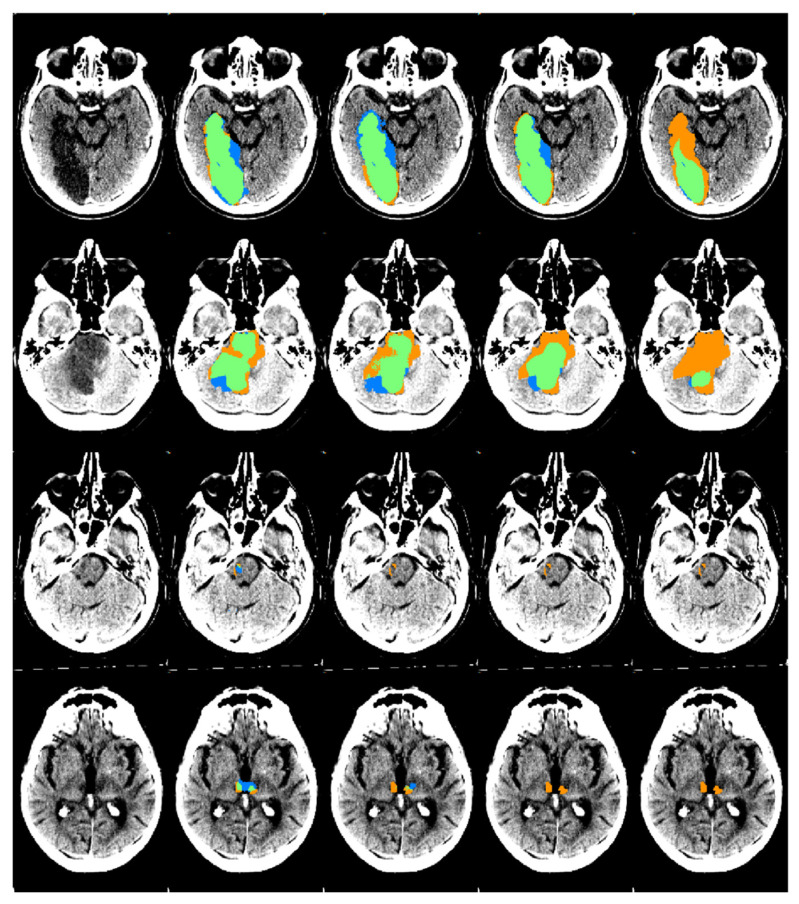
An example of automatic segmentation results obtained by the four CNNs on the PCS test set. From the left to the right column: the original scan, the automatic segmentation results from the TL-CNN, PCS-CNN, CD-CNN, and ACS-CNN are shown. The segmentation maps show true positives (green), false positives (blue), and false negatives (orange). The scans were plotted using a window center around 35, with a window width of 30.

**Table 1 diagnostics-11-01621-t001:** Baseline characteristics, treatment and time data for patients with posterior circulation stroke and anterior circulation stroke. Prior to posterior stroke, transient ischemic attack (TIA), posterior circulation TIA, and atrial fibrillation (history or 12 lead electrocardiogram (ECG)) were not available (NAV) for the HERMES dataset. We use abbreviations for the National Institutes of Health Stroke Scale (NIHSS), modified Ranking Scale (mRS) and, Intravenous Thrombolysis (IVT).

Parameter	Posterior Stroke	Anterior Stroke
Clinical		
Age, years, mean (Standard Deviation)	65.65 (12.2)	66.1 (13.3)
Sex, F, No. [%]	34/107 [31.8]	458/1018 [45]
NIHSS at baseline, mean [median] (N)	21.4 [19] (107)	17 [17] (1015)
Prior Conditions		
Diabetes mellitus, No. [%]	28/107 [26.2]	169/1018 [16.6]
Hypertension, No. [%]	64/107 [59.8]	564/1018 [55.4]
Stroke, No. [%]	21/107 [19.6]	121/1018 [11.9]
Posterior circulation stroke, No. [%]	7/107 [6.5]	NAV
TIA, No. [%]	10/106 [9.4]	NAV
Posterior circulation TIA, No. [%]	2/106 [1.9]	NAV
Atrial fibrillation, No. [%]	13/107 [12.1]	314/1018 [30.8]
Atrial fibrillation (history or 12 lead ECG), No. [%]	23/107 [21.5]	NAV
Pre-Stroke mRS		
0, No. [%]	80/107 [74.8]	836/1017 [82.1]
1, No. [%]	11/107 [10.3]	129/1017 [12.7]
2, No. [%]	13/107 [12.1]	29/1017 [2.9]
3, No. [%]	3/107 [2.8]	23/1017 [2.3]
Treatment		
IVT, No. [%]	92/107 [86]	872/1018 [85.7]
Time		
Stroke onset to IVT, min., mean [Standard Deviation] (N)	176.9 [176.102] (90)	112.2 [57.2] (871)

## Data Availability

The data used in this study are available on reasonable request from the corresponding author. The data are not publicly available due to patients’ privacy.

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
