# Peer review of "Automated Final Lesion Segmentation in Posterior Circulation Acute Ischemic Stroke Using Deep Learning"

_diagnostics, 2021, doi:10.3390/diagnostics11091621_

Round 1

Reviewer 1 Report

The article developed by Riaan Zoetmulder et al. is well prepared/developed and can be accepted for publication. Methods and discussion are balanced for the message to be conveyed.
Correct the ERROR information at the bottom of page 6.

Decrease de size of Figure 2.

Edit all Tables based on MDPI rules.

Author Response

Thank you for your review. We apologize for these mistakes. We have processed your feedback; the size of Figure 2 has been decreased to fit the rest of the text, the error information has been corrected, and the tables have been edited based on MDPI rules.

Reviewer 2 Report

For measuring FLV, PCS was automatically segmented with CNN.

Major points

According to Table 2, dice coefficients are too low in four CNNs. This hinders the clinical application of authors’ CNN.

Compared with the ref 6, dice coefficients are low in the current study.

https://www.ncbi.nlm.nih.gov/pmc/articles/PMC7476369/

“A lesion was defined as detected if the percentage of overlapping voxels between the automatic and reference segmentations was larger than a predefined threshold.” “To account for this issue the required percentage of overlapping voxels was set to either greater than zero percent or a more conservative 20%. “ The definition of predefined threshold is ambiguous in this paper.

Dice and ICC must be calculated for each case of test set between manual segmentations by two radiologists.

For FLV measurement, detection rate is not a primary metric.

Please cite the following papers for segmentation with transfer learning.

https://www.ajronline.org/doi/abs/10.2214/AJR.19.22347

https://www.frontiersin.org/articles/10.3389/frai.2021.694815/full

https://onlinelibrary.wiley.com/doi/pdf/10.1002/mrm.27969

“Error! Reference source not found” These must be edited.

Minor points

FLV and FU-NCCT must be defined precisely in main text.

“the input block” Size of input must be clarified in the section 2.4.

Please refer to Appendix in main text.

“33.    Yosinski, J.; Clune, J.; Bengio, Y.; Lipson, H. How transferable are features in deep neural networks? 2014.” Journal name?

Round 2

Reviewer 2 Report

“The latter measure, lesion volume, is also clinically a more relevant measure than the spatial overlap.“ If so, results of Bland Altman plots shown in Figure 1 must be evaluated clinically. Is the difference of volume between ground truth and prediction acceptable clinically? Please discuss about Limits of Agreement of this study.
